# Simple Steps to Success: A Method for Step-Based Counterfactual Explanations

**Jenny Hamer**[*]                                                           *hamer@google.com*
*Google DeepMind, New York*

**Nicholas Perello**[*]                                                      *nperello@umass.edu*
*University of Massachusetts, Amherst*

**Jason Valladares**[†]                                                 *jason.valladares3@gmail.com*
*Google*

**Vignesh Viswanathan**[*]                                              *vviswanathan@umass.edu*
*University of Massachusetts, Amherst*

**Yair Zick**                                                                *yzick@umass.edu*
*University of Massachusetts, Amherst*

**Reviewed on OpenReview:** *https://openreview.net/forum?id=R6ey5DKaoX*

## Abstract

Algorithmic recourse is a process that leverages counterfactual explanations, going beyond understanding why a system produced a given classification, to providing a user with actions they can take to change their predicted outcome. Existing approaches to compute such interventions—known as *recourse*—identify a set of points that satisfy some desiderata—e.g. an intervention in the underlying causal graph, minimizing a cost function, etc. Satisfying these criteria, however, requires extensive knowledge of the underlying model structure, an often unrealistic amount of information in several domains. We propose a data-driven and model-agnostic framework to compute counterfactual explanations. We introduce StEP, a computationally efficient method that offers *incremental steps* along the data manifold that directs users towards their desired outcome. We show that StEP uniquely satisfies a desirable set of axioms. Furthermore, via a thorough empirical and theoretical investigation, we show that StEP offers provable robustness and privacy guarantees while outperforming popular methods along important metrics.

## 1 Introduction

An automatic decision maker produces a negative prediction for some user—e.g. denies their grad school application, offers them bad loan terms or an overly strict criminal sentence; what can the user do to change this outcome? Counterfactual explanations (Wachter et al., 2017) recommend actions that change algorithmic predictions on a given point. This is usually modeled as a constrained optimization problem that outputs specific points which satisfy certain desirable properties: actionability, validity, data manifold closeness and causality to name a few (Verma et al., 2020). Achieving these desiderata often requires both significant compute power and user/model information. We propose a **lightweight algorithm for producing counterfactual explanations**; rather than searching for good interventions for users, we search for good *directions* that users can take. We use these directions to create an iterative recourse mechanism where stakeholders can repeatedly request new directions after carrying out the recommended changes. We

---

[*]equal contribution
[†]work completed during Master's at the University of Massachusetts, Amherst

show that by carefully choosing these directions, we satisfy several desirable properties of algorithmic recourse at a significantly lower computational cost.

## 1.1 Our Contributions

We propose a recourse algorithm called *Stepwise Explainable Paths* (StEP). Our key theoretical insight is that StEP is the *only method* for generating recourse directions (counterfactual explanations) which satisfies a set of natural properties. StEP directions are model-agnostic and easy to compute, requiring only the training dataset and the output of the model of interest on points in the training dataset. That is, our method does not require prior knowledge of the underlying model architecture and instead takes a strongly data-dependent approach. In addition to introducing StEP, a novel step-based recourse method, we present its provable quality (Section 3.1), diversity (Section 3.2), and privacy (Section 3.3) guarantees. We also provide an extensive experimental evaluation of StEP, including a holistic cross-comparison with three popular recourse methods (DiCE (Mothilal et al., 2020a), FACE (Poyiadzi et al., 2020), and C-CHVAE (Pawelczyk et al., 2020)) on three widely-used financial datasets—Credit Card Default (Yeh & Lien, 2009), Give Me Some Credit (Credit Fusion, 2011) and UCI Adult (Kohavi, 1996) datasets. We also investigate StEP's robustness to noise (Section 4.3).

## 1.2 Related Work

Counterfactual explanations are a fundamental concept in the model explanation literature (Verma et al., 2020). First proposed by Wachter et al. (2017), they are founded in legal interpretations of explainability (Wachter et al., 2018) and are distinct from *feature highlighting* methods (Barocas et al., 2020), such as Shapley value-based methods (Datta et al., 2016; Lundberg & Lee, 2017), Local Interpretable Model-Agnostic Explanations (Ribeiro et al., 2016) and saliency maps (Simonyan et al., 2013). Recent policy efforts describe algorithmic recourse as a means for fostering trust in AI systems (Wachter et al., 2018; NTIA, 2023; Biden Jr., 2023). Whereas feature highlighting methods indicate important features (or feature interactions (Patel et al., 2021)), counterfactual explanations identify *changes* to features that are likely to change the outcome(Karimi et al., 2022; Verma et al., 2020). These changes are usually solutions to an optimization problem, ensuring that the explanations are valid, actionable, sparse and diverse (Mothilal et al., 2020a; Karimi et al., 2020a; Ustun et al., 2019a). These solutions are computed using integer linear programs (Ustun et al., 2019a; Kanamori et al., 2020), SAT solvers (Karimi et al., 2020a), or gradient descent (Mothilal et al., 2020a; Wachter et al., 2017). Other works provide a sequence of steps from the point of interest to a point with the desirable outcome along the data manifold (Poyiadzi et al., 2020). While these prior methods offer reasonable approaches to recourse generation, none of them axiomatically characterize their approach.

Another approach in the literature is to solve the causal problem of finding the best intervention (Karimi et al., 2021). Karimi et al. (2021) argue that any recourse recommendation must be consistent with the underlying causal relations between variables. However, this requires complete (or, as in Karimi et al. (2020b), imperfect) knowledge of the underlying causal model, which is often practically or computationally infeasible.

We address this issue by providing users with promising directional actions, allowing them more flexibility and agency in enacting recourse recommendations. Our axiomatic characterization is similar to that of Monotone Influence Measures (MIM) (Sliwinski et al., 2019). However, Sliwinski et al.'s approach is not iterative in nature, nor is their recommended direction guaranteed to change the prediction outcome.

## 2 Preliminaries

We introduce some general notation and definitions used throughout this work. We denote by $\vec{x} \in \mathbb{R}^m$ a *point of interest* (PoI) corresponding to an individual or their current state, and define a dataset $\mathcal{X} = \{\vec{x}^1, \vec{x}^2, \dots, \vec{x}^n\} \subseteq \mathbb{R}^m$ with $m$ features and $n$ datapoints. We use $x_i$ to denote the $i$-th index of the vector $\vec{x}$.

We focus on binary classification using a *model of interest* $f : \mathbb{R}^m \mapsto \pm 1$ trained on the dataset $\mathcal{X}$; our goal is to produce a counterfactual explanation for $\vec{x} \notin \mathcal{X}$ where $f(\vec{x}) = -1$. The *counterfactual explanation* for a point of interest $\vec{x}$ is a direction (or a set of directions) $\vec{d} \in \mathbb{R}^m$ that moves the point $\vec{x}$ towards the positive

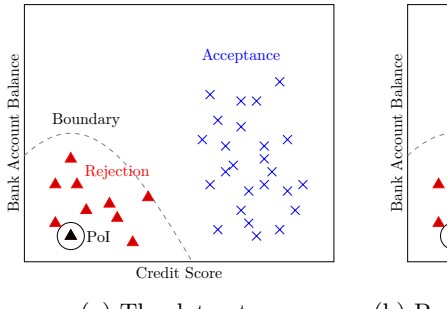 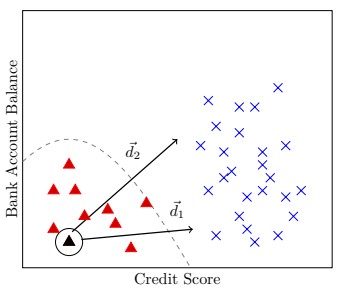 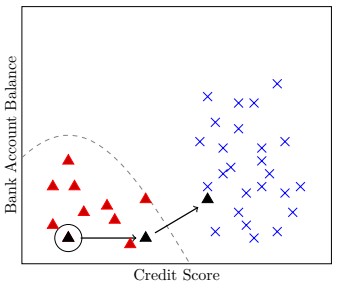

(a) The dataset       (b) Possible Recourse Directions       (c) Possible Stakeholder Path

Figure 1: (Left) The training dataset that the loan approval algorithm uses, (Middle) Plotting possible recourse directions for Example 2.1 and (Right) plotting a stakeholder path toward loan acceptance.

class, i.e. $f(\vec{x} + c \cdot \vec{d}) = 1$ for some positive value $c > 0$. We go beyond simply providing an explanation for the the PoI's outcome by framing our output as recommended *recourse* which can be actioned by the user. Upon applying recourse to a PoI $\vec{x}$, we refer to $\vec{x} + c \cdot \vec{d}$ as a *counterfactual* (CF) of $\vec{x}$ denoted by $\vec{x}_{CF}$.

Offering a single explanation may not be helpful in real-world settings; users may be unable to make a single dramatic change to their features or may not exactly follow the suggested recourse. We allow the stakeholder to repeatedly request new recourse directions as they change their values until they are positively classified. We call such an approach *direction-based recourse*. To illustrate this approach, consider the following example of algorithmic loan approval.

**Example 2.1.** Consider a loan applicant whose application is rejected by an algorithm utilizing two factors: bank account balance and credit score. The loan applicant, i.e., the PoI, the algorithm, and the training dataset are described in Figure 1a. Several directions can change the PoI's value from $\triangle$ (red points with label $-1$) to $\times$ (blue points with label $+1$). For example, increasing the credit score while leaving the bank account balance unchanged eventually changes the applicant's label; increasing both bank balance and credit score (Figure 1b) results in a positive outcome. Suppose that we provide the applicant with both alternatives; the applicant makes some modifications and reapplies for a loan. If the application is rejected, we provide them with new directions, based on their current state, and repeat until the application is accepted. This results in a recourse *path*, rather than a single direction (Figure 1c).

We wish to ensure that our directions are robust; thus, even if the stakeholder somewhat deviates from our suggested path, they are still likely to secure a positive outcome. This also offloads some of the computational costs onto the stakeholder, which guarantees certain recourse properties that would otherwise require a significant amount of information and computational cost. Consider Example 2.1: after providing two different directions, the change that the applicant makes to their datapoint will likely be a minimum cost change that approximately follows one of the directions we propose in Figure 1b. Even if the resulting change does not facilitate a change in outcome, or is incorrectly executed, we can still offer additional directions until a desirable outcome is obtained.

## 2.1 Data-driven Recourse

Whether explanations should be computed based on the underlying model or the observed data is a highly debated topic (Chen et al., 2020; Barocas et al., 2020; Janzing et al., 2020). While data-driven recourse methods exist (Poyiadzi et al., 2020), most work solely with the model of interest $f$ (Mothilal et al., 2020a; Wachter et al., 2017; Karimi et al., 2020a).

Recent work shows that explanations can perform poorly when they are inconsistent with the data manifold—i.e. the underlying distribution from which the data is drawn (Frye et al., 2021; Aas et al., 2021) and are vulnerable to manipulation (Slack et al., 2020; 2021). Ignoring the data manifold when computing counterfactuals can be even more pernicious, resulting in recourse recommendations that may not improve the outcome in any way, but simply move it outside the data manifold. To see why this is the case, consider

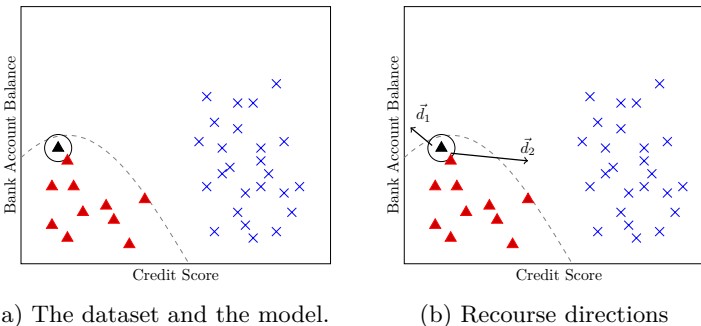

(a) The dataset and the model.  (b) Recourse directions

Figure 2: (Left) The training dataset that the loan approval algorithm uses, and (Right) Off manifold $(\vec{d_1})$ and On manifold $(\vec{d_2})$ directions.

---

**Algorithm 1** Stepwise Explainable Paths (StEP)

---

**Require:** Dataset $\mathcal{X}$ partitioned into $k$ clusters $\{\mathcal{X}_1, \ldots, \mathcal{X}_k\}$, point of interest $\vec{x}$, model $f$, some non-negatively valued function $\alpha : \mathbb{R}_{\geq 0} \mapsto \mathbb{R}_{\geq 0}$
1: **while** $f(\vec{x}) = -1$ **do**
2:     **for** every cluster $c \in [k]$ **do**           ▷ Generate a direction $\vec{d_c}$ for each $c$
3:         $\vec{d_c} \leftarrow \sum_{\vec{x}' \in \mathcal{X}_c} (\vec{x}' - \vec{x}) \alpha(\|\vec{x}' - \vec{x}\|) \mathbb{1}(f(\vec{x}') = 1)$
4:     **end for**
5:     Offer the directions $\{d_c\}_{c \in [k]}$ to the stakeholder
6:     Stakeholder returns an updated point of interest $\vec{x}^*$
7:     $\vec{x} \leftarrow \vec{x}^*$
8: **end while**

---

a simplified, two dimensional instance as shown in Example 2.1 but with a different point of interest (given in Figure 2a). The shortest distance perturbation which crosses the decision boundary corresponds to the direction $\vec{d_1}$ (as given in Figure 2b). This direction suggests that the stakeholder decrease their credit score and marginally increase their bank account balance, which actively moves the stakeholder away from the positively classified points. An explanation algorithm that recognizes the data manifold will offer a direction similar to $\vec{d_2}$, arguably making the stakeholder a better candidate for loan approval.

## 3 Stepwise Explainable Paths (StEP)

We now present our approach for computing recourse directions. The overall method is summarized in Algorithm 1. We first partition the dataset $\mathcal{X}$ into $k$ clusters $\{\mathcal{X}_1, \ldots, \mathcal{X}_k\}$ (using a standard clustering algorithm). For a point of interest $\vec{x}$, we generate a direction $\vec{d_c}$ towards each cluster $X_c$ using the expression

$$\vec{d_c} = \sum_{\vec{x}' \in \mathcal{X}_c} (\vec{x}' - \vec{x}) \alpha(\|\vec{x}' - \vec{x}\|) \mathbb{1}(f(\vec{x}') = 1) \tag{1}$$

where $\alpha : \mathbb{R}_+ \to \mathbb{R}_+$ is some non-negative function and $\|.\|$ is a rotation invariant distance metric. We select directions using Equation (1) (a similar formula is proposed by Sliwinski et al. (2019)). The intuition behind this equation is as follows: for each point $\vec{x}'$ in the cluster $\mathcal{X}_c$, if $f(\vec{x}') = 1$, we 'move' on the line $(\vec{x} - \vec{x}')$ a distance of $\alpha(\|\vec{x}' - \vec{x}\|)$, where $\alpha$ is a decreasing function of $\|\vec{x} - \vec{x}'\|$. Thus, points closer to $\vec{x}$ have a greater effect on $\vec{d_c}$; similarly, positively classified points that are close to each other will have a greater effect on $\vec{d_c}$. We offer these $k$ directions to the stakeholder, who then returns with a new point $\vec{x}'$ after following the recourse recommendations. The process is repeated until we produce a positively classified point (i.e. a counterfactual) or reach a user-specified maximum number of iterations. StEP's computational complexity is linear in the size of the dataset and is bounded by the maximum number of iterations.

### 3.1 An Axiomatically Justified Direction

We axiomatically derive our choice of direction in equation 1: we identify a direction that uniquely satisfies a set of desirable properties. More specifically, for a given point of interest $\vec{x} \notin \mathcal{X}_c$, we believe any reasonable recourse direction, denoted by $\vec{d}(\vec{x}, \mathcal{X}_c, f)$ should satisfy the following axioms:

**Shift Invariance (SI)** Let $\mathcal{X}_c + \vec{b}$ denote the dataset resulting from adding the vector $\vec{b}$ to each point in $\mathcal{X}$ and let $f_{\vec{b}}$ be a shifted model of interest such that $f_{\vec{b}}(\vec{z}) = f(\vec{z} - \vec{b})$ for all $\vec{z}$. Then, $\vec{d}(\vec{x}, \mathcal{X}_c, f) = \vec{d}(\vec{x} + \vec{b}, \mathcal{X}_c + \vec{b}, f_{\vec{b}})$.

**Rotation/Reflection Faithfulness (RRF)** Let $A$ be any matrix with $det(A) \in \{-1, +1\}$ and let $A\mathcal{X}_c$ denote the dataset resulting from replacing every point $\vec{x}^j$ in $\mathcal{X}_c$ with $A\vec{x}^j$. Let $f_A$ denote a rotated model of interest such that $f_A(\vec{z}) = f(A^{-1}\vec{z})$ for all $\vec{z}$. Then, $A\vec{d}(\vec{x}, \mathcal{X}_c, f) = \vec{d}(A\vec{x}, A\mathcal{X}_c, f_A)$.

**Continuity (C)** $\vec{d}$ is a continuous function of the dataset $\mathcal{X}_c$. One can think of the continuity axiom as a notion of recourse *robustness*: small changes to the input PoI will not cause significant changes to the resulting recourse.

**Data Manifold Symmetry (DMS)** Let $f$ and $g$ be two functions such that $f(\vec{x}^j) = g(\vec{x}^j)$ for all points $\vec{x}^j \in \mathcal{X}$. Then, we must have $\vec{d}(\vec{x}, \mathcal{X}_c, f) = \vec{d}(\vec{x}, \mathcal{X}_c, g)$.

**Negative Classification Indifference (NCI)** Let $\vec{x}' \in \mathcal{X}_c$ be a datapoint with $f(\vec{x}') = -1$. Then, $\vec{d}(\vec{x}, \mathcal{X}_c, f) = \vec{d}(\vec{x}, \mathcal{X}_c \setminus \{\vec{x}'\}, f)$.

**Positive Classification Monotonicity (PCM)** Let $\vec{x}' \notin \mathcal{X}_c$ be a point with $f(\vec{x}') = 1$ and $x_i' > x_i$, then $\vec{d}(\vec{x}, \mathcal{X}_c, f) \le \vec{d}_i(\vec{x}, \mathcal{X}_c \cup \{\vec{x}'\}, f)$. Similarly, if $x_i' < x_i$, then $\vec{d}_i(\vec{x}, \mathcal{X}_c, f) \ge \vec{d}_i(\vec{x}, \mathcal{X}_c \cup \{\vec{x}'\}, f)$.

Our axioms are inspired by Sliwinski et al. (2019), who use a similar approach to characterize a family of direction-based explanations (*Monotone Influence Measures*). Unlike the MIM framework, we remove any dependency on negatively classified points (the Negative Classification Indifference axiom). Without this change, a naive adaptation of MIM may output directions pointing away from all positively classified points. Intuitively, a cluster of negative points near positive points may make it impossible to recommend any recourse, as the MIM framework "shies away" from negative point clusters (see details in Appendix A).

The five remaining axioms are fundamental. Shift Invariance and Rotation/Reflection Faithfulness ensure the directions depend on the relative locations of the points rather than their absolute values. The RRF axiom also generalizes the *feature symmetry* axiom: swapping the coordinates of features $i$ and $j$ does not change the value assigned to them; this is commonly used in the characterization of other model explanation frameworks (Datta et al., 2015; 2016; Patel et al., 2021; Lundberg & Lee, 2017; Sliwinski et al., 2019). In addition, these properties ensure the units in which we measure feature values have no effect on the outcome, e.g., measuring income in dollars rather than cents has no effect on the importance of income. Continuity ensures that the direction we pick is robust to small changes in the cluster $\mathcal{X}_c$. Data Manifold Symmetry (DMS) ensures that the direction depends on the model of interest only through points in the dataset. In other words, DMS ensures that the model's output on points outside the data manifold do not affect the output direction — a desirable property in model explanations (Frye et al., 2021; Lundberg & Lee, 2017; Chen et al., 2020). Positive Classification Monotonicity (PCM) ensures that the direction will point towards regions with a large number of positively classified points. In other words, if there is a large number of positively classified points with a high value in some feature $i \in N$ (e.g. bank balance), PCM ensures that the output direction will ask the stakeholder to increase their bank balance.

Any direction which satisfies the above axioms is uniquely given by Equation (1): i.e. it is a weighted combination of the directions from the point of interest to every positively classified point in the dataset. The weight given to every point is a function of their distance $\|\vec{x}' - \vec{x}\|$.

**Theorem 3.1.** *A recourse direction for a point of interest $\vec{x}$ given a dataset $\mathcal{X}_c$, a model of interest $f$ and a rotation invariant distance metric $\|.\|$ satisfies SI, RRF, C, DMS, NCI and PCM if and only if it is given by equation 1.*

*Proof.* For readability, we replace $\mathcal{X}_c$ with $\mathcal{X}$. It is easy to see that equation 1 satisfies all five axioms so we only show uniqueness.

We assume without loss of generality that $\mathcal{X}$ contains no negatively classified points. If $\mathcal{X}$ does contain any negatively classified points, we can simply remove them without changing the recourse direction because of Negative Classification Indifference (NCI). Therefore, our goal is to show that any direction which satisfies (SI), (RRF), (C), (DMS), (NCI) and (PCM) is of the form

$$\vec{d}(\vec{x}, \mathcal{X}, f) = \sum_{\vec{x}' \in \mathcal{X}} \alpha(\|\vec{x}' - \vec{x}\|)(\vec{x}' - \vec{x})$$

We start off with a useful lemma.

**Lemma 3.2.** *If any direction $\vec{d}$ satisfies Rotation and Reflection Faithfulness (RRF) and Positive Classification Monotonicity (PCM), then for any dataset $\mathcal{X}$, any datapoint $\vec{x}$, any model of interest $f$ and any positively classified point $\vec{y} \neq \vec{x}$, there exists some $a \geq 0$ such that*

$$\vec{d}(\vec{x}, \mathcal{X} \cup \{\vec{y}\}, f) - \vec{d}(\vec{x}, \mathcal{X}, f) = a(\vec{y} - \vec{x})$$

*Proof.* Suppose for contradiction that there is some $\vec{x}$, $\mathcal{X}$, $\vec{y}$ and $f$ such that

$$\forall a \geq 0 : \vec{d}(\vec{x}, \mathcal{X} \cup \{y\}, f) - \vec{d}(\vec{x}, \mathcal{X}, f) \neq a(\vec{y} - \vec{x})$$

Let $\vec{l} = \vec{d}(\vec{x}, \mathcal{X} \cup \{y\}, f) - \vec{d}(\vec{x}, \mathcal{X}, f)$. Let $A$ be a rotation matrix such that $(Al)_1 < 0$ and $[A(\vec{y} - \vec{x})]_1 > 0$; such a matrix exists since the two vectors are either linearly independent or $\vec{l} = -b(\vec{y} - \vec{x})$ where $b \in \mathbb{R}^+$. Since $\vec{d}$ satisfies Rotation and Reflection Faithfulness (RRF), we have from $(Al)_1 < 0$

$$\vec{d}_1(A\vec{x}, A\mathcal{X} \cup \{A\vec{y}\}, f_A) - \vec{d}_1(A\vec{x}, A\mathcal{X}, f_A) < 0$$

This contradicts the Positive Classification Monotonicity (PCM) property since $(A\vec{y})_1 > (A\vec{x})_1$ and $f_A(A\vec{y}) = f(\vec{y}) = 1$. $\qquad\square$

Consider a direction $\vec{d}$ that satisfies the six desired axioms. We go ahead and show uniqueness via induction on $|\mathcal{X}|$. Let $k = 0$, $\mathcal{X} = \{\}$. By Shift Invariance (SI), $\vec{d}(\vec{x}, \{\}, f) = \vec{d}(\vec{0}, \{\}, f_{-\vec{x}})$. The vector $\vec{0}$ and an empty $\mathcal{X}$ are invariant under rotation. Therefore, since $\vec{d}$ satisfies Rotation and Reflection Faithfulness (RRF) and Data Manifold Symmetry (DMS), we must have $\vec{d}(\vec{0}, \{\}, f_{-\vec{x}}) = \vec{0}$, the only vector invariant under rotation.

Let $k = 1$, $\mathcal{X} = \{\vec{y}\}$ where $\vec{y} \neq \vec{x}$. Note that any pair of $(\vec{x}, \vec{y})$ can be translated by shift and rotation to any other pair $(\vec{x}', \vec{y}')$ with the same distance ($\|\vec{y} - \vec{x}\|$) between them. This is because the distance metric ($\|.\|$) is rotation invariant and any distance metric is shift invariant when computing the distance between two points; the shifts cancel each other out. Note that after rotation by some matrix $A$, we have $f_A(A\vec{y}) = f(\vec{y})$ and similarly, after shift by some vector $\vec{b}$, we have $f_{\vec{b}}(\vec{y} + \vec{b}) = f(\vec{y})$. Therefore, the label of the point $y$ does not change after applying Shift Invariance (SI) or Rotation and Reflection Faithfulness (RRF). Therefore, by (SI), (RRF) and Lemma 3.2, we have

$$\vec{d}(\vec{x}, \mathcal{X}, f) = \alpha(\|\vec{y} - \vec{x}\|)(\vec{y} - \vec{x})$$

where $\alpha$ is a non-negative valued function.

Suppose the hypothesis holds when $|\mathcal{X}| \leq k$. Consider a dataset $\mathcal{Y}$ of size $k + 1$. This means $\mathcal{Y}$ contains at least two distinct points $\vec{y}, \vec{z} \neq \vec{x}$. We prove our hypothesis for the case where $\vec{y}$ and $\vec{z}$ are linearly independent. The case where they are linearly dependent follows from Continuity (C): we can peturb the vectors slightly to make them linearly independent. By Lemma 3.2, we have

$$\vec{d}(\vec{x}, \mathcal{Y}, f) \in A = \{\vec{d}(\vec{x}, \mathcal{Y} \setminus \{\vec{y}\}, f) + a(\vec{y} - \vec{x})\}$$
$$\text{and } \vec{d}(\vec{x}, \mathcal{Y}, f) \in B = \{\vec{d}(\vec{x}, \mathcal{Y} \setminus \{\vec{z}\}, f) + a(\vec{z} - \vec{x})\} \tag{2}$$

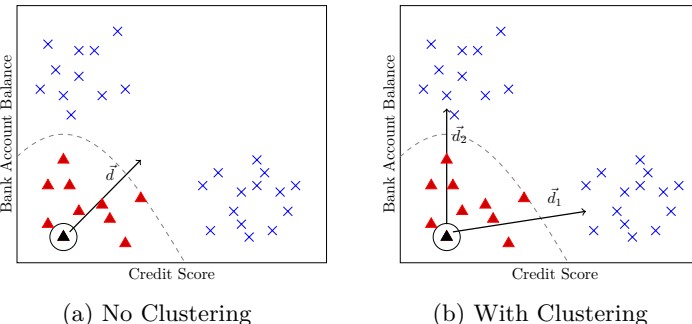

Figure 3: Directly applying the direction formula equation 1 without clustering to find recourse directions may result in undesirable behavior, e.g. excluding a variety of options and picking an off-manifold direction (Left). Clustering resolves this issue (Right).

By the inductive hypothesis, we have

$$\vec{d}(\vec{x}, \mathcal{Y} \setminus \{\vec{y}\}, f) = \vec{d}(\vec{x}, \mathcal{Y} \setminus \{\vec{y}, \vec{z}\}, f) + \alpha(\|\vec{z} - \vec{x}\|)(\vec{z} - \vec{x})$$
$$\text{and } \vec{d}(\vec{x}, \mathcal{Y} \setminus \{\vec{z}\}, f) = \vec{d}(\vec{x}, \mathcal{Y} \setminus \{\vec{y}, \vec{z}\}, f) + \alpha(\|\vec{y} - \vec{x}\|)(\vec{y} - \vec{x})$$

We can use this and the fact that $\vec{y} - \vec{x}$ and $\vec{z} - \vec{x}$ are linearly independent to combine the two sets in equation 2 to get

$$\vec{d}(\vec{x}, \mathcal{Y}, f) = A \cap B = \vec{d}(\vec{x}, \mathcal{Y} \setminus \{\vec{y}, \vec{z}\}, f) + \alpha(\|\vec{y} - \vec{x}\|)(\vec{y} - \vec{x}) + \alpha(\|\vec{z} - \vec{x}\|)(\vec{z} - \vec{x})$$

This completes the induction. □

Reducing stakeholder "travel distance" to reach a positive classification is a common objective used in algorithmic recourse (Mothilal et al., 2020a; Karimi et al., 2020a; Mahajan et al., 2019). This distance can be seen as a measure of the effort required from the stakeholder to change their outcome. We can incorporate this in StEP using the choice of $\alpha$. Any $\alpha$ function of the form $\alpha(z) = \frac{1}{z^k}$ with $k \geq 2$ ensures that nearby points are assigned a greater weight than far-off points. Different choices of $\alpha$ result in different directions.

### 3.2 Diverse and On-Manifold Recourse via Clustering

Despite its theoretical soundness, utilizing StEP without clustering presents two drawbacks.

**Lack of guaranteed diversity** In many cases, changing the $\alpha$ function does not result in a diverse set of directions even when many directions are possible. Consider a slight modification to Example 2.1 (given by Figure 3) where the class of positive points is separated into two clusters. A recourse algorithm could reasonably output a direction towards either cluster as a potential recourse. However, Equation (1) outputs a linear combination of these two clusters. Furthermore, it is easy to see that changing the function $\alpha$ will not significantly change the direction.

**Off-manifold directions** Since Equation equation 1 aggregates directions, it is possible to obtain off-manifold directions (refer to Figure 3a); off-manifold regions may have high prediction error.

Without clustering, Equation equation 1 aggregates directions obtained for different parts of the dataset, rather than treating different data regions differently. We resolve both aforementioned concerns by *clustering* the positively classified data points and compute directions per individual cluster. This ensures that we identify different clusters of points and aggregate each cluster in a theoretically sound manner. The impacts are demonstrated in Figure 3.

### 3.3 Privacy-Preserving Direction Selection

One potential concern when using the dataset directly to compute recourse is that StEP could potentially leak sensitive user data (indeed, other model explanation algorithms have been shown to leak private information

Milli et al. (2019); Shokri et al. (2021)). With clustering, it may not be possible to offer privacy guarantees since the clustering process itself may not be privacy preserving. However, we can show that the StEP distance computation itself is privacy-preserving (see Dwork & Roth (2014) for a formal exposition to differential privacy). Briefly, an algorithm $\mathcal{M}$ that takes as input a dataset $\mathcal{X}$ and outputs a value $\mathcal{M}(\mathcal{X})$ is said to be $(\epsilon, \delta)$-differentially private if for all $S \subseteq \text{range}(\mathcal{M})$, we have:

$$\Pr[\mathcal{M}(\mathcal{X}) \in S] \leq e^{\epsilon} \Pr[\mathcal{M}(\mathcal{X}') \in S] + \delta$$

where $\mathcal{X}$ and $\mathcal{X}'$ are any two datasets that differ by at most one datapoint. Differential privacy states that the output of $\mathcal{M}$ does not vary much by the removal of any data point. A simple method to prove that a function is differentially private is to upper bound its *sensitivity*, i.e. the change to the function when adding a datapoint. Dwork & Roth (2014) show that adding a finite amount of noise to a bounded sensitivity function guarantees differential privacy. A similar guarantee is offered for Shapley-based (Datta et al., 2016), and gradient-based (Patel et al., 2022) explanations.

StEP (without clustering) can be made differentially private for a specific family of $\alpha$ functions, when the distance metric used is the $\ell_2$ norm. More formally, if we assume that $\alpha(z) \leq \frac{C}{z}$ for all $z > 0$, then we can bound the sensitivity of StEP.

**Lemma 3.3.** *When the distance metric used is the $\ell_2$ norm and $\alpha(z) \leq \frac{C}{z}$ for all $z > 0$, the global sensitivity (using the $\ell_2$ norm) of the direction given by equation 1 is upper bounded by $C$.*

*Proof.* Let the direction output of equation 1 (in the absence of clustering) for a particular dataset $\mathcal{X}$, a model of interest $f$ and a point of interest $\vec{x}$ be $\vec{d}(\vec{x}, \mathcal{X}, f)$.

The global sensitivity of equation 1 using the $l_2$ norm is given by

$$\Delta_2 \vec{d} = \max_{\vec{x}, \mathcal{X}, \mathcal{X}'} \|\vec{d}(\vec{x}, \mathcal{X}, f) - \vec{d}(\vec{x}, \mathcal{X}', f)\|_2 \tag{3}$$

where $\mathcal{X}'$ is $\mathcal{X}$ with one additional (or one less) datapoint (Dwork & Roth, 2014). We can assume without loss of generality that $\mathcal{X}'$ contains one additional positively classified datapoint $\vec{x}'$. If the point is not positively classified, then none of the directions change and the sensitivity is 0. The global sensitivity defined in equation 3 reduces to

$$\Delta_2 \vec{d} = \max_{\vec{x}, \vec{x}'} \|(\vec{x}' - \vec{x})\alpha(\|\vec{x}' - \vec{x}\|_2)\mathbb{I}(f(\vec{x}') = 1)\|_2 \quad \leq \max_{\vec{x}, \vec{x}'} \left\|(\vec{x}' - \vec{x})\frac{C}{\|\vec{x}' - \vec{x}\|_2}\right\|_2 \text{(by assumptions on } \alpha \text{ and } f(\vec{x}'))$$

$$\leq \max_{\vec{x}, \vec{x}'} \frac{1}{\|\vec{x}' - \vec{x}\|_2}\|C(\vec{x}' - \vec{x})\|_2 = C$$

In the first inequality, we assume $\|\vec{x} - \vec{x}'\|_2 > 0$ so we can apply $\alpha(\|\vec{x} - \vec{x}'\|_2) \leq \frac{C}{\|\vec{x} - \vec{x}'\|_2}$. If $\|\vec{x} - \vec{x}'\|_2 = 0$, then by definition we must have $\vec{x} = \vec{x}'$ which implies $\Delta_2 \vec{d} = 0$ and the lemma trivially holds. $\square$

Since the direction has bounded sensitivity, classic results from the differential privacy literature tell us that introducing Gaussian noise makes the direction $(\epsilon, \delta)$ differentially private.

**Theorem 3.4.** *When the distance metric used is the $\ell_2$ norm and $\alpha(z) \leq \frac{C}{z}$, the directions output by equation 1 can be made $(\epsilon, \delta)$-differentially private by adding Gaussian noise with $0$ mean and standard deviation $\sigma \geq \frac{\beta C^2}{\epsilon}$ where $\beta^2 > 2\log(\frac{1.25}{\delta})$ to all the features.*

The proof of Theorem 3.4 is a direct application of Dwork & Roth (2014, Theorem 3.22) and is omitted. Offering multiple recourse directions results in an additive increase in privacy cost. More specifically, if we provide $k$ directions, and each direction is $(\epsilon, \delta)$ differentially private, then our mechanism is $(k\epsilon, k\delta)$ differentially private (Dwork & Roth, 2014).

| Dataset | Type | Classification task | LogReg | RandForest | DNN |
|---------|------|---------------------|--------|------------|-----|
| Credit Card Default | Financial | payment default | 1000 | 535 | 559 |
| Give Me Some Credit | Financial | financial distress | 1000 | 332 | 426 |
| UCI Adult | Demographic | income >$50k | 1000 | 1000 | 1000 |

Table 1: Dataset descriptions. Values for logistic regression (LogReg), random forest (RandForest) and 2-layer DNN (DNN) reflect average number of negatively classified datapoints in the test set across 10 trials. We limit the number of datapoints used for recourse (from the test split) to 1000.

## 4 Empirical Evaluation & Analysis

We compare the performance of StEP and three popular recourse methods—DiCE (Mothilal et al., 2020a), FACE (Poyiadzi et al., 2020) and C-CHVAE (Pawelczyk et al., 2020)—on three widely used datasets within counterfactual research using three base models. We also examine StEP's robustness to noise or "user-interference" when following recourse directions. We provide an overview of our experimental setup here and include full details to support reproducibility in Appendix C. Recall that a counterfactual point for $\vec{x}$ is a terminal point of a recourse path, $\vec{x}_{CF}$, such that $f(\vec{x}_{CF}) = 1$.

**Recourse Baselines** Given a negatively classified PoI $\vec{x}$, DiCE solves an optimization problem that outputs a diverse set of counterfactuals. For each of these counterfactual points $\vec{x}_{CF}$, $(\vec{x}_{CF} - \vec{x})$ can be interpreted as a direction recommendation for the $\vec{x}$. FACE constructs an undirected graph over the set of datapoints and finds a path from the point of interest $\vec{x}$ to a set of positively classified *candidate* points using Djikstra's algorithm (Dijkstra, 1959). Each edge in the path that connects $\vec{x}$ to $\vec{x}_{CF}$ can be thought of as a direction recommendation from $\vec{x}$ to $\vec{x}_{CF}$. The backend of C-CHVAE (Pawelczyk et al., 2020) is a variational autoencoder (VAE): distances within the latent space surrounding the PoI is used to identify counterfactual points. We use the author's implementation for DiCE and adapt FACE's and C-CHVAE's implementations from Poyiadzi et al. (2020) and Pawelczyk et al. (2021).

Given a negatively classified PoI $\vec{x}$, StEP and FACE produce a sequence of points $(\vec{x}^0, \vec{x}^1, \ldots, \vec{x}^\ell)$ where $\vec{x}^0$ is the original PoI, $\vec{x}^1$ is the point after following the first direction recommendation by the recourse method, and so on. DiCE and C-CHVAE produce two point sequences $(\vec{x}^0, \vec{x}^\ell)$. We refer to this sequence of points as a *recourse path*, and each of the directions can be referred to as a "step".

**Datasets and Models** We employ three real-world datasets in our cross-comparison analysis: Credit Card Default (Yeh & Lien, 2009), Give Me Some Credit (Credit Fusion, 2011), and UCI Adult/Census Income (Kohavi, 1996), described in Table 1. For each dataset, we train and validate **logistic regression**, **random forest**, and **two-layer DNN** model instances following a 70/15/15 training, validation, and test (recourse-time) data splits. Based on a balanced hyperparameter tuning across all recourse methods, we specify a confidence threshold of 0.7 at test time for each base model to determine whether a PoI is positively classified. Implementation and hyperparameter tuning details are described in Appendix C.2.

**Metrics & Properties** In alignment with counterfactual and recourse literature, we use the following well-established metrics to evaluate the performance of StEP, DiCE, FACE, and C-CHVAE on each base model and dataset (Verma et al., 2020). In the following definitions, we are given a recourse path $(\vec{x}^0, \vec{x}^1, \ldots, \vec{x}^\ell)$, where $\vec{x}^0$ is the PoI $\vec{x}$. A recourse path is *successful* if $f(\vec{x}^\ell) = 1$, i.e. the recommended path ultimately produced a counterfactual $\vec{x}^\ell = \vec{x}_{CF}$. Our results report the average of these metrics over all PoIs.

**Success** (or *validity* (Verma et al., 2020)) measures the proportion of PoIs with a successful path.

**Average Success** measures the proportion of successful paths generated for a given PoI $\vec{x}$.

$\ell_2$ **Distance** is the Euclidean distance between the PoI $\vec{x}^0$ and the final point $\vec{x}^\ell$, i.e. $\|\vec{x}^0 - \vec{x}^\ell\|_2$. Distance is only computed for successful paths. Low distance, or *proximal*, recourse, supports actionability.

**Diversity** is the average Euclidean distances between the counterfactuals of each successful recourse path for a given PoI $\vec{x}$. Diversity is only computed for PoIs with least two successful recourse paths.

| Dataset | Method | Logistic Regression | | | | Random Forest | | | | DNN | | | | Max Error % |
|---|---|---|---|---|---|---|---|---|---|---|---|---|---|---|
| | | Success | Avg Success | $\ell_2$ Dist. | Diversity | Success | Avg Success | $\ell_2$ Dist. | Diversity | Success | Avg Success | $\ell_2$ Dist. | Diversity | |
| Credit Card Default | StEP | 1.00 | 0.91 | 7.06 | 2.58 | 1.00 | 0.84 | 3.20 | 0.95 | 1.00 | 1.00 | 5.04 | 1.29 | 8.51 |
| | DiCE | 1.00 | 1.00 | 35.28 | 12.93 | 1.00 | 1.00 | 20.34 | 8.35 | 0.99 | 0.99 | 32.47 | 13.02 | 2.96 |
| | FACE | 0.54 | 0.54 | 4.46 | 0.78 | 0.51 | 0.51 | 2.75 | 0.88 | 0.45 | 0.45 | 4.38 | 0.93 | 3.16 |
| | CCHVAE | 1.00 | 1.00 | 7.88 | 1.17 | 1.00 | 1.00 | 3.22 | 0.29 | 1.00 | 1.00 | 5.11 | 0.34 | 5.03 |
| Give Me Some Credit | StEP | 0.98 | 0.70 | 15.32 | 10.73 | 1.00 | 0.74 | 4.03 | 9.91 | 1.00 | 0.99 | 5.87 | 2.15 | 15.94 |
| | DiCE | 0.99 | 0.99 | 103.68 | 38.77 | 1.00 | 1.00 | 73.54 | 31.16 | 0.99 | 0.99 | 95.45 | 37.14 | 3.33 |
| | FACE | 0.98 | 0.98 | 2.97 | 0.57 | 0.96 | 0.96 | 2.79 | 0.63 | 0.93 | 0.93 | 2.80 | 0.59 | 0.98 |
| | CCHVAE | 0.06 | 0.06 | 2.24 | 0.13 | 1.00 | 1.00 | 1.40 | 0.02 | 1.00 | 1.00 | 4.18 | 0.03 | 15.07 |
| UCI Adult | StEP | 1.00 | 0.54 | 2.39 | 1.37 | 0.89 | 0.47 | 4.93 | 2.09 | 1.00 | 0.56 | 2.40 | 1.38 | 4.01 |
| | DiCE | 1.00 | 1.00 | 6.75 | 1.76 | 1.00 | 1.00 | 7.57 | 1.35 | 1.00 | 1.00 | 7.03 | 1.60 | 5.66 |
| | FACE | 0.63 | 0.63 | 3.02 | 0.74 | 0.63 | 0.63 | 2.77 | 0.77 | 0.64 | 0.64 | 3.02 | 0.76 | 0.83 |
| | CCHVAE | 1.00 | 1.00 | 2.57 | 0.46 | 0.93 | 0.92 | 2.59 | 0.44 | 0.99 | 0.99 | 2.77 | 0.51 | 8.48 |
| Max Error % | | 6.70 | 7.25 | 15.94 | 15.07 | 1.89 | 2.95 | 7.86 | 6.49 | 1.50 | 1.50 | 7.72 | 14.83 | |

Table 2: Comparative analysis results on all datasets and base models. Metrics are computed on scaled data and averaged over 10 trials. We include max. standard error bounds for each metric by task and across tasks.

## 4.1 Categorical Variables and Actionability

We provide rigorous support for categorical variables, including *immutable*, *semi-mutable*, *ordinal* and *unordered*. Immutable features—ones that cannot be changed, e.g. `race`—are ignored during recourse. Semi-mutable features can only be changed in one direction, e.g., education level can only increase. Ordinal features, e.g. an ordered scale like ratings, are encoded features using a Borda scale (from 1 to $k$). The remaining categorical features are encoded via one-hot encoding.

For recourse to be practically relevant, methods should offer *actionable* directions (Ustun et al., 2019b): those which do not suggest infeasible changes or changes to immutable features, i.e. that the user can actually execute. *Constraints* are a natural way of preventing recommendations that ask a user to perform an infeasible action, e.g. becoming younger or less educated to secure a loan. StEP encodes such constraints within its distance metric, in a manner similar to DiCE (Mothilal et al., 2020a). For each dataset and task, we implement all appropriate encodings and constraints so that the directions produced by StEP satisfy actionability desiderata and requirements. Refer to Appendix C.1 for encodings by dataset.

## 4.2 Comparative Analysis of StEP, DiCE, FACE, and C-CHVAE

For each PoI, we generate $k = 3$ recourse paths, repeat this over 10 trials, and compute metrics and statistics based on the resulting counterfactuals. The comparison of StEP, DiCE, FACE, and C-CHVAE on all base models are presented in Table 2. We introduce additional metrics and further discussion in Appendix B.

**StEP** For all tasks, StEP offers a balance between minimizing distance and maximizing diversity performance. StEP's lower distance recourse supports actionability—more proximal counterfactuals may be more actionable to the user—while providing variety in the suggested directions. We observe differences between distance and diversity—while maintaining a desirable trade-off between the two metrics—across base models while consistently performing well on success. On UCI Adult, StEP exhibits significantly different success compared to average success. Given that StEP produces a recourse path for each of its underlying $k$ clusters, average success may be reduced when one or more of these clusters contains many outliers. However, even in this setting, StEP produces successful counterfactuals for the remaining clusters. This supports StEP's robustness to outliers and generalization to the tail of the underlying data distribution.

StEP is also robust to the choice of base model, providing a consistent balance between proximal and diverse counterfactuals with high success. Even with a simple base model (logistic regression), StEP balances desirable metrics, while reducing $\ell_2$ distance under more complex yet lightweight models (Random Forest or DNN).

**DiCE** Across tasks, DiCE consistently produces successful recourse paths. Since DiCE selects counterfactuals by solving an optimization problem with a diversity objective, we unsurprisingly observe that the method consistently excels on this metric. This performance comes with a trade-off of much higher distance from the given PoI. This suggests that DiCE could be useful when highly varied recourse options are desirable, but at the cost of potentially inactionable recourse given the very high distance to the PoI (and therefore very significant changes to be made by the user).

**FACE** When FACE produces successful recourse, it consistently performs well w.r.t. $\ell_2$ distance—in other words, its counterfactuals are close to the original PoI. In aggregate across datasets, however, FACE's sensitivity to distance between datapoints in its underlying graph results in unreliable success performance. Our results suggest that increasing the nonlinearity of the base model does not assist FACE in this dimension. Additionally, FACE exhibits a significant trade-off between low distance between a PoI and its counterfactual(s) and diversity. For $k > 1$, FACE finds the closest counterfactual, an objective which is orthogonal to promoting diversity.

**C-CHVAE** Across tasks, C-CHVAE performs competitively on $\ell_2$ distance, but produces counterfactuals with very low diversity. C-CHVAE's objective relies on similarity via distance in the latent space of its VAE, so each of the successful $k$ recourse paths produces a counterfactual which is as close as possible to the original PoI. Across nearly all tasks, C-CHVAE excels in terms of success but exhibits a trade-off between proximal successful counterfactuals and diversity.

Despite these strengths, we observe a core weakness of C-CHVAE demonstrated by its surprisingly poor success on Give Me Some Credit under the logistic regression base model. To ensure that generated counterfactuals lie on the data manifold, for a given PoI, C-CHVAE's autoencoder approximates the conditional log likelihood of its mutable features given the immutable features. Therefore, for a given immutable value (e.g. a protected class like race or gender), if there are few datapoints with high-confidence positive predictions, C-CHVAE will generate counterfactuals based on low-confidence predictions. In this task using logistic regression, only $8\%$ of positively classified training examples had a feature value $age \leq 59$. This significant feature imbalance means that C-CHVAE generated counterfactuals with a confidence level of $< 0.7$ for these datapoints, demonstrating C-CHVAE's sensitivity to feature imbalance and risk in magnifying biased data.

Unlike with StEP, when DiCE, FACE, and C-CHVAE are unsuccessful at producing recourse for a given PoI, they fail for all $k$ paths, resulting in equal success and average success.

### 4.3 Practical Robustness Considerations for StEP

**Clustering.** We evaluate StEP's sensitivity to the choice of underlying clusters across all tasks and base models. Firstly, using off-the-shelf *k-means*, we vary $k$ between $\{1, \ldots, 6\}$. Our empirical results in Appendix B.6 show that StEP is robust to both the number and relative size of clusters when using $k$-means. We also evaluate StEP under **random clustering**, by assigning each point to a random cluster value from $\{1, \ldots, k\}$. Our metrics of interest slightly improve as $k$ increases. We hypothesize that this improvement stems from the finer-grained dataset partitioning, which allows StEP to produce a path (within the given maximum number of iterations) from the PoI to at least one cluster. Even under random clustering, StEP performs well on most metrics. Given the strict categorical constraints in the UCI Adult task, the center of each random cluster being distributed uniformly across the entire dataset results in some clusters being inherently less reachable for many PoIs.

**User interference.** Algorithmic recourse models commonly assume that users exactly follow suggested actions. In Appendix B.7, we relax this assumption by introducing a noise parameter $\beta$ as a proxy for how much a user deviates from the prescribed direction. We construct a noise vector where each dimension that represents a continuous feature in the dataset is independently sampled from the standard normal distribution

| Dataset | Noise ($\beta$) | Logistic Regression | | | | Random Forest | | | | DNN | | | | Max Error % |
|---|---|---|---|---|---|---|---|---|---|---|---|---|---|---|
| | | Success | Avg Success | $\ell_2$ Dist. | Diversity | Success | Avg Success | $\ell_2$ Dist. | Diversity | Success | Avg Success | $\ell_2$ Dist. | Diversity | |
| Credit Card Default | 0.0 | 1.00 | 0.91 | 7.06 | 2.58 | 1.00 | 0.84 | 3.20 | 0.95 | 1.00 | 1.00 | 5.04 | 1.29 | 8.51 |
| | 0.1 | 1.00 | 0.90 | 7.09 | 2.58 | 1.00 | 0.87 | 3.22 | 0.95 | 1.00 | 1.00 | 5.04 | 1.29 | 8.51 |
| | 0.3 | 1.00 | 0.92 | 7.10 | 2.57 | 1.00 | 0.90 | 3.20 | 0.96 | 1.00 | 1.00 | 5.02 | 1.30 | 8.54 |
| | 0.5 | 1.00 | 0.95 | 7.06 | 2.57 | 1.00 | 0.91 | 3.13 | 0.97 | 1.00 | 1.00 | 5.01 | 1.31 | 8.64 |
| Give Me Some Credit | 0.0 | 0.98 | 0.70 | 15.32 | 10.73 | 1.00 | 0.74 | 4.03 | 9.91 | 1.00 | 0.99 | 5.87 | 2.15 | 15.94 |
| | 0.1 | 0.99 | 0.81 | 13.68 | 9.98 | 1.00 | 1.00 | 3.50 | 2.14 | 1.00 | 0.99 | 5.87 | 2.15 | 14.90 |
| | 0.3 | 0.99 | 0.90 | 9.95 | 7.73 | 1.00 | 1.00 | 2.83 | 1.69 | 0.96 | 0.95 | 3.90 | 2.15 | 14.33 |
| | 0.5 | 0.99 | 0.92 | 8.03 | 6.37 | 1.00 | 1.00 | 2.42 | 1.44 | 0.95 | 0.95 | 3.45 | 2.10 | 13.09 |
| UCI Adult | 0.0 | 1.00 | 0.54 | 2.39 | 1.37 | 0.89 | 0.47 | 4.93 | 2.09 | 1.00 | 0.56 | 2.40 | 1.38 | 4.01 |
| | 0.1 | 1.00 | 0.53 | 2.35 | 1.36 | 0.89 | 0.47 | 4.93 | 2.07 | 1.00 | 0.55 | 2.35 | 1.37 | 3.84 |
| | 0.3 | 1.00 | 0.53 | 2.34 | 1.36 | 0.89 | 0.47 | 4.91 | 2.07 | 1.00 | 0.55 | 2.34 | 1.37 | 3.94 |
| | 0.5 | 1.00 | 0.53 | 2.32 | 1.35 | 0.89 | 0.47 | 4.86 | 2.06 | 1.00 | 0.55 | 2.31 | 1.36 | 3.96 |
| **Max Error %** | | 1.06 | 7.25 | 15.94 | 12.41 | 1.64 | 2.95 | 11.14 | 14.24 | 0.72 | 0.80 | 9.41 | 14.90 | |

Table 3: User-interference experiment results. Metrics are computed on scaled data and averaged over 10 trials. Maximum standard error bounds for each metric by task and across tasks are included.

and the remaining dimensions are zero. We scale this noise vector to magnitude $\beta \times \|\vec{d}\|$, where $\vec{d}$ is the original recourse vector and $\beta \in \mathbb{R}^m_{\geq 0}$. The noise vector is then added to $\vec{d}$ as the next suggested action.

The results presented in Table 3 demonstrate StEP's robustness to user-interference and other noise. The *improvements* in performance w.r.t. success, avg success, and $\ell_2$ distance in many cases can be interpreted as noise providing a benefit similar to small amounts of stochasticity in gradient methods (e.g. SGD, momentum), helping StEP move out of a local minima. We include additional results and discussion on this task in Appendix B.7. This case study provides considerable empirical evidence to StEP's robustness to deviation from the suggested recourse.

## 5 Discussion and Conclusion

We introduce StEP, a data-driven method for direction-based algorithmic recourse that does not depend on the underlying model or knowledge of its underlying causal relations. We show that the directions computed by StEP uniquely satisfy a set of desirable properties, which can be made differentially private under some mild assumptions. We empirically demonstrate StEP's ability to produce actionable and diverse recourse, its robustness to user-interference, and its practical utility.

In Section 3.2, we discuss the limitations of StEP without clustering and highlight the necessity of a clustering-based approach. Modulating $k$ controls the number of clusters (and, in turn, number of recourse directions produced); a clustering with too fine of a granularity may result in unsuccessful recourse paths for some of the clusters. While standard clustering methods perform well in practice, a thorough analysis of their effects on StEP could provide useful insights in practical deployment settings.

An additional limitation of StEP is its dependence on how we measure distances between datapoints, as with DiCE, FACE and Wachter's method. Defining new encoding schemas for categorical variables and investigating different notions of distance in the use of StEP and other recourse methods is an interesting area of future work. Finally, StEP and other recourse methods raise questions regarding group fairness, e.g. whether StEP offers similar performance for users from different protected groups.

**Acknowledgements.** The authors thank Neel Patel for early discussions. Perello and Zick are supported by Army Research Lab DEVCOM Data and Analysis Center - Contract W911QX23D0009.

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

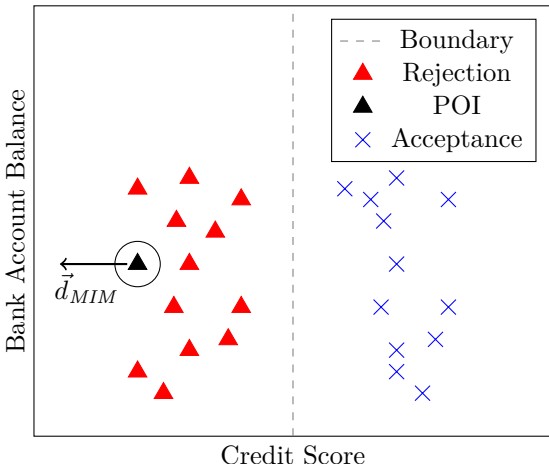

Figure 4: An example where the MIM direction points away from all the positively classified points.

## A  Issue with the Naive Adaptation of MIM

MIM (proposed by Sliwinski et al. (2019)) is a data driven approach to explain the outcome of point $\vec{x}$. Intuitively, it is a direction that moves towards points of the same outcome and moves away from points of the opposite outcome. It is given by:

$$\vec{d}(\vec{x}, \mathcal{X}, f) = \sum_{\vec{x}' \in \mathcal{X}} (\vec{x}' - \vec{x}) \alpha(\|\vec{x}' - \vec{x}\|) \mathbb{1}(f(\vec{x}') = f(\vec{x}))$$

where $\alpha$ is a non-negative valued function and $\mathbb{1}(.)$ takes the value 1 if the input condition is true and 0 otherwise. We can naively adapt MIM for recourse by flipping the direction, moving towards positively classified points and away from negatively classified points. This would give us the direction:

$$\vec{d}(\vec{x}, \mathcal{X}, f) = \sum_{\vec{x}' \in \mathcal{X}} (\vec{x}' - \vec{x}) \alpha(\|\vec{x}' - \vec{x}\|) \mathbb{1}(f(\vec{x}') \neq f(\vec{x}))$$

This direction, unfortunately, can lead to bad recourse recommendations. To see why, consider Example 2.1 with a different dataset and point of interest (given in Figure 4). When nearby points are given a higher weight than far away points, the direction output by MIM could be poor, pointing away from all the positively classified points. This is because when nearby points are given a higher weight than far away points, the push away from negatively classified points can be stronger than the pull towards positively classified points. This is indeed what happens in Figure 4.

## B  Additional Empirical Investigation

We compare StEP with three methods, DiCE, FACE, and C-CHVAE by generating $k = 3$ recourse instructions for each negatively classified datapoint in the test set for each dataset.

With StEP, we first partition the positively labeled training data into 3 clusters, and then for each PoI in the test set, we produce a direction for each of the 3 clusters. In these comparative experiments, we use sci-kit learn's default k-means implementation without any tuning, and we assume the the stakeholder follows the provided direction exactly.

### B.1  Baseline Recourse Methods

We compare StEP to DiCE (Mothilal et al., 2020a), FACE (Poyiadzi et al., 2020), and C-CHVAE Pawelczyk et al. (2020). DiCE outputs a diverse set of points, FACE outputs a set of paths that terminate at a positively classified point, and C-CHVAE outputs points in the latent neighborhood of the PoI using an autoencoder.

Given a point $\vec{x}$ such that $f(\vec{x}) = -1$, DiCE uses determinantal point processes and solves the following optimization problem to output a diverse set of $m$ counterfactual explanations $\{\vec{c}_1, \vec{c}_2, \ldots, \vec{c}_m\}$:

$$\underset{\vec{c}_1, \vec{c}_2, \ldots, \vec{c}_m}{\arg\min} \; \frac{1}{m} \sum_{i=1}^{m} loss(f(\vec{c}_i), 1) + \frac{\lambda_1}{m} \sum_{i=1}^{m} dist(\vec{c}_i, \vec{x})$$
$$-\lambda_2 \text{dpp\_diversity}(\vec{c}_1, \vec{c}_2, \ldots, \vec{c}_m)$$

Each of the output points $\vec{c}_j$ can be seen as a recommendation for the direction $\vec{c}_j - \vec{x}$. Due to the structure of the optimization problem, none of these output points are guaranteed to be positively classified. We run DiCE with the default hyperparameter settings.

FACE constructs an undirected graph over the set of data points and finds a path from the point of interest to a set of positively classified *candidate* points using Djikstra's algorithm (Dijkstra, 1959). All the results generated in this paper use a distance threshold of 3 and the max length of a path is capped at 50 data points.

C-CHVAE utilizes a variational autoencoder to construct a low-dimensional neighborhood within a radius around the PoI and searches for counterfactuals within it. At each iteration the radius to search around gets expanded by a given step distance hyperparameter. We run C-CHVAE with a step distance of 1 and set the max number of iteration to 50. For the remaining hyperparameters, we use the default given by Pawelczyk et al. (2021). We modify Pawelczyk et al.'s implementation for C-CHVAE to support non-binary categorical features and $k > 1$ counterfactuals.

## B.2  Additional Comparative Analysis

We define three appropriate metrics, *path length*, *path steps*, and *proximal diversity* motivated by our direction-based algorithm.

**Path length** is the sum of Euclidean distances between the steps of a recourse path. A shorter path length requires stakeholders to work less to change their outcome. Formally, the path length is given by $\sum_{i=1}^{\ell} \|\vec{x}^i - \vec{x}^{i-1}\|_2$. Path lengths are only computed for successful paths.

**Path steps** corresponds to the number of steps taken in the path, given by $\ell - 1$. When the number of steps is 1 (as is with DiCE and C-CHVAE), distance and path length are the same.

**Proximal Diversity** (Prox. Diver.) is the total Euclidean distances between the counterfactual points of each successful recourse path for a given PoI $\vec{x}$ normalized by distance from the furthest terminal point to the PoI,

$$\frac{1}{\max_{i \in 1, \ldots, k} \|\vec{x}^0 - \vec{x}_i^\ell\|_2} \sum_{i=1}^{k} \sum_{j=i}^{k} \|\vec{x}_i^\ell - \vec{x}_j^\ell\|_2.$$

To provide diverse recourse one can simply provide counterfactual points significantly distant from each other and the PoI. This makes recourse less actionable and the metric value less interpretable, therefore we normalize by max distance (proximity) between the PoI and its counterfactuals to scale this. Proximal diversity is only computed for PoIs where at least two recourse paths are successful.

The results for these new metrics are presented in Table 4. Proximal diversity reflects our observations on distance and diversity in Section 4.2. That is, StEP has diverse and close in distance recourse and DiCE has highly diverse and far in distance recourse. Proximal diversity weighs these two aspects, resulting in similar performance between the two methods. Given that DiCE and C-CHVAE do not output points between the PoI and counterfactual, path steps was always one step.

## B.3  Discussion of DiCE

Since DiCE is not a direction-based method, we generate $k$ counterfactual points and interpret the difference between the original PoI $\vec{x}$ and counterfactual $\vec{x}_{CF}$ as a proxy for the path to take. In our experiments, DiCE almost always generates a valid counterfactual explanation but this counterfactual is usually much

| Dataset | Method | Logistic Regression | | | | Random Forest | | | | DNN | | | | Max Error % |
| | | Success | Path Length | Path Steps | Prox. Diver. | Success | Path Length | Path Steps | Prox. Diver. | Success | Path Length | Path Steps | Prox. Diver. | |
|---|---|---|---|---|---|---|---|---|---|---|---|---|---|---|
| Credit Card Default | StEP | 1.00 | 7.23 | 7.42 | 2.49 | 1.00 | 3.70 | 3.75 | 2.22 | 1.00 | 5.04 | 5.13 | 2.09 | 6.88 |
| | DiCE | 1.00 | 35.28 | 1.00 | 2.43 | 1.00 | 20.34 | 1.00 | 2.23 | 0.99 | 32.47 | 1.00 | 2.64 | 2.96 |
| | FACE | 0.54 | 5.61 | 2.18 | 2.02 | 0.51 | 3.68 | 1.56 | 2.49 | 0.45 | 5.04 | 2.06 | 2.16 | 1.75 |
| | CCHVAE | 1.00 | 7.88 | 1.00 | 0.96 | 1.00 | 3.22 | 1.00 | 0.76 | 1.00 | 5.11 | 1.00 | 0.57 | 3.53 |
| Give Me Some Credit | StEP | 0.98 | 15.86 | 15.86 | 2.41 | 1.00 | 4.12 | 4.12 | 2.09 | 1.00 | 5.87 | 5.87 | 2.19 | 15.42 |
| | DiCE | 0.99 | 103.68 | 1.00 | 2.34 | 1.00 | 73.54 | 1.00 | 2.28 | 0.99 | 95.45 | 1.00 | 2.38 | 3.33 |
| | FACE | 0.98 | 2.72 | 1.07 | 1.92 | 0.96 | 2.80 | 1.01 | 2.01 | 0.93 | 2.81 | 1.01 | 1.91 | 1.13 |
| | CCHVAE | 0.06 | 2.24 | 1.00 | 0.13 | 1.00 | 1.40 | 1.00 | 0.15 | 1.00 | 4.18 | 1.00 | 0.13 | 15.07 |
| UCI Adult | StEP | 1.00 | 2.70 | 2.70 | 2.48 | 0.89 | 5.35 | 5.46 | 2.46 | 1.00 | 2.57 | 2.65 | 2.47 | 3.98 |
| | DiCE | 1.00 | 6.75 | 1.00 | 1.65 | 1.00 | 7.57 | 1.00 | 1.20 | 1.00 | 7.03 | 1.00 | 1.43 | 3.98 |
| | FACE | 0.63 | 4.29 | 1.87 | 2.14 | 0.63 | 3.83 | 1.72 | 2.41 | 0.64 | 4.18 | 1.84 | 2.16 | 0.83 |
| | CCHVAE | 1.00 | 2.57 | 1.00 | 1.24 | 0.93 | 2.69 | 1.00 | 1.24 | 0.99 | 2.83 | 1.00 | 1.35 | 5.19 |
| Max Error % | | 6.70 | 15.42 | 15.42 | 15.07 | 2.21 | 7.58 | 7.58 | 5.13 | 1.50 | 7.72 | 7.72 | 6.88 | |

Table 4: Comparative analysis results on all datasets and base models. Metrics are computed on scaled data and averaged over 10 trials. The maximum standard error bounds for each metric by task (row) and across tasks (column) are included.

farther from the PoI than StEP on average, which requires the user to make very large changes at each step of recourse. Furthermore, sometimes the counterfactual can be away from the data manifold.

## B.4 Discussion of FACE

FACE is a direction-based recourse method which finds the shortest path from each PoI $\vec{x}$ to $k$ positively classified candidate points that exist within the graph (i.e. counterfactuals). The existence of edges between points in a graph are determined using a *distance threshold* parameter. If two points have a distance less than the distance threshold, an edge is generated; otherwise, no edge spans the two points. Intuitively, the distance threshold determines the size of the step users are required to take when following FACE's recourse.

Finding an appropriate distance threshold for FACE is a challenging and highly task-dependent requirement. If the distance threshold is too small, the graph becomes sparse and typically produces few to no successful recourse paths. On the other hand, if the distance threshold is very large, the graph becomes dense and recourse generates trivial paths between PoIs and candidate points consisting of a single edge. In this case, FACE produces a brittle recourse.

One possible option for the distance threshold is to set it equal to the step-size of StEP (equal to 1) to ensure a fair comparison of the two methods, which results in very few successful paths being generated (in some tasks, with a success rate as low as 0). When increasing the distance threshold between 2 to 3, FACE begins non-trivial paths across our tasks. All the results generated in this paper are using the distance threshold 3.

## B.5 Discussion of C-CHVAE

C-CHVAE uses both the encoder and decoder from a variational autoencoder trained on the given dataset. The encoder transforms data points into a low-dimensional representation and the decoder reconstructs latent representations to the original dimension. During it's search, C-CHVAE encodes the PoI and samples points around the hyper-sphere around it. These points are then decoded and tested on the base model to determine if they are counterfactuals. In each iteration, C-CHVAE expands the hyper-sphere it samples around by a given step distance hyperparamter. To find $k$ counterfactuals, we continue the algorithm until $k$ counterfactuals are found via sampling.

| Dataset | Noise ($\beta$) | Logistic Regression | | | | Random Forest | | | | DNN | | | | Max Error % |
|---|---|---|---|---|---|---|---|---|---|---|---|---|---|---|
| | | Success | Path Length | Path Steps | Prox. Diver. | Success | Path Length | Path Steps | Prox. Diver. | Success | Path Length | Path Steps | Prox. Diver. | |
| Credit Card Default | 0.0 | 1.00 | 7.23 | 7.42 | 2.49 | 1.00 | 3.70 | 3.75 | 2.22 | 1.00 | 5.04 | 5.13 | 2.09 | 6.88 |
| | 0.1 | 1.00 | 7.43 | 7.73 | 2.49 | 1.00 | 4.26 | 4.31 | 2.22 | 1.00 | 5.06 | 5.15 | 2.09 | 6.82 |
| | 0.3 | 1.00 | 7.91 | 8.34 | 2.49 | 1.00 | 4.75 | 4.85 | 2.25 | 1.00 | 5.19 | 5.33 | 2.09 | 6.66 |
| | 0.5 | 1.00 | 8.36 | 8.93 | 2.48 | 1.00 | 4.73 | 4.92 | 2.28 | 1.00 | 5.44 | 5.69 | 2.11 | 6.41 |
| Give Me Some Credit | 0.0 | 0.98 | 15.86 | 15.86 | 2.41 | 1.00 | 4.12 | 4.12 | 2.09 | 1.00 | 5.87 | 5.87 | 2.19 | 15.42 |
| | 0.1 | 0.99 | 16.26 | 16.27 | 2.42 | 1.00 | 3.52 | 3.53 | 2.77 | 1.00 | 5.89 | 5.89 | 2.20 | 11.20 |
| | 0.3 | 0.99 | 12.59 | 12.64 | 2.48 | 1.00 | 2.92 | 2.93 | 2.86 | 0.96 | 4.03 | 4.05 | 2.24 | 9.71 |
| | 0.5 | 0.99 | 10.45 | 10.57 | 2.54 | 1.00 | 2.60 | 2.62 | 2.94 | 0.95 | 3.77 | 3.81 | 2.29 | 8.52 |
| UCI Adult | 0.0 | 1.00 | 2.70 | 2.70 | 2.48 | 0.89 | 5.35 | 5.46 | 2.46 | 1.00 | 2.57 | 2.65 | 2.47 | 3.98 |
| | 0.1 | 1.00 | 2.64 | 2.94 | 2.49 | 0.89 | 5.81 | 6.34 | 2.47 | 1.00 | 2.46 | 2.82 | 2.47 | 4.76 |
| | 0.3 | 1.00 | 2.63 | 3.01 | 2.49 | 0.89 | 5.79 | 6.47 | 2.46 | 1.00 | 2.44 | 2.88 | 2.47 | 4.72 |
| | 0.5 | 1.00 | 2.62 | 3.16 | 2.48 | 0.89 | 5.70 | 6.73 | 2.45 | 1.00 | 2.43 | 3.02 | 2.47 | 4.77 |
| Max Error % | | 1.06 | 15.42 | 15.42 | 5.20 | 1.64 | 11.20 | 11.20 | 2.35 | 0.72 | 9.41 | 9.41 | 6.88 | |

Table 5: User-interference analysis results on all datasets and base models. Metrics are computed on scaled data and averaged over 10 trials. The maximum standard error bounds for each metric by task (row) and across tasks (column) are included. "Prox. Diver." is shorthand for proximal diversity.

In most tasks, this iterative process of expanding the search radius step-by-step aids C-CHVAE in producing low distance counterfactuals with high success. However, the encoding and decoding steps lead C-CHVAE to be sensitive to strong relationships between mutable and immutable features, leading to failures in certain tasks as described in Section 4.2. The choice of number of iterations, step distance, or number of $k$ counterfactuals did not affect the poor success on Give Me Some Credit under logistic regression. We recommend careful dataset selection when using C-CHVAE, as the relation between immutable and mutable features can be and is often discriminatory.

## B.6 The Effects of Clustering on StEP

StEP supports base clustering methods which allow the user to specify the number of clusters (e.g. $k$-means) and those which estimate a number of clusters (e.g. affinity propagation). With the latter, the clustering method determines $k$, the number of potential recourse paths StEP can produce.

We suggest using clustering approaches where the user can specify the number of clusters (number of recourse paths), aligning with more traditional recourse methods. Methods that empirically determine a number of clusters to satisfy some objective function can result in a large $k$–while having multiple recourse paths is desirable, past a certain threshold loses its marginal utility, can overload users, and become challenging to interpret *if one is considering all of those paths without additional post-processing.*

The consideration around number of clusters has more to do with how the clusters are distributed in high-dimensional space relative to the learned decision boundary of each classifier, rather than as a function of "cluster goodness" or quality. From these results, we show that StEP is robust to the number of clusters for k-means, and therefore also robust to the relative size of each cluster.

We expect that clustering can negatively impact recourse when the clustering method and base model leverages immutable features. Intuitively, as an example within the Give Me Some Credit, one of the k clusters reflects an older/retired population unlikely to experience financial distress. Additionally, as observed in the C-CHVAE analysis in Section 4.2, only 8% of positively classified training examples had a feature value *age* $\leq$ 59 under logistic regression. Therefore, StEP consistently fails to find a path to this cluster for PoIs younger than 59 years old. This is reflected in StEP 's Avg Success under logistic regression and Give

| Dataset | $k$ | Logistic Regression | | | | Random Forest | | | | DNN | | | | Max Error % |
|---|---|---|---|---|---|---|---|---|---|---|---|---|---|---|
| | | Success | Avg Success | $\ell_2$ Dist. | Diversity | Success | Avg Success | $\ell_2$ Dist. | Diversity | Success | Avg Success | $\ell_2$ Dist. | Diversity | |
| Credit Card Default | 1 | 0.90 | 0.90 | 5.03 | 0.00 | 1.00 | 1.00 | 2.99 | 0.00 | 1.00 | 1.00 | 4.65 | 0.00 | 1.27 |
| | 2 | 0.92 | 0.87 | 6.17 | 2.01 | 1.00 | 1.00 | 3.10 | 0.77 | 1.00 | 1.00 | 4.95 | 1.10 | 1.46 |
| | 3 | 1.00 | 0.91 | 7.06 | 2.58 | 1.00 | 0.84 | 3.20 | 0.95 | 1.00 | 1.00 | 5.04 | 1.29 | 8.51 |
| | 4 | 1.00 | 0.93 | 6.96 | 4.48 | 1.00 | 0.86 | 3.30 | 1.78 | 1.00 | 1.00 | 5.06 | 2.38 | 6.54 |
| | 5 | 1.00 | 0.95 | 7.06 | 4.96 | 1.00 | 0.88 | 3.65 | 2.52 | 1.00 | 1.00 | 5.19 | 2.83 | 11.13 |
| | 6 | 1.00 | 0.95 | 7.22 | 5.50 | 1.00 | 0.90 | 3.44 | 2.40 | 1.00 | 1.00 | 5.29 | 3.37 | 8.56 |
| Give Me Some Credit | 1 | 0.58 | 0.58 | 2.37 | 0.00 | 1.00 | 1.00 | 2.83 | 0.00 | 1.00 | 1.00 | 4.11 | 0.00 | 8.67 |
| | 2 | 0.90 | 0.70 | 8.78 | 6.58 | 1.00 | 0.73 | 4.12 | 10.28 | 1.00 | 1.00 | 4.17 | 0.27 | 31.71 |
| | 3 | 0.98 | 0.70 | 15.32 | 10.73 | 1.00 | 0.74 | 4.03 | 9.91 | 1.00 | 0.99 | 5.87 | 2.15 | 15.94 |
| | 4 | 1.00 | 0.75 | 15.27 | 18.64 | 1.00 | 0.79 | 6.56 | 16.35 | 1.00 | 0.98 | 6.26 | 4.42 | 10.51 |
| | 5 | 1.00 | 0.73 | 15.60 | 19.85 | 1.00 | 0.81 | 7.21 | 17.57 | 1.00 | 0.98 | 6.11 | 4.66 | 11.00 |
| | 6 | 1.00 | 0.73 | 14.73 | 19.11 | 1.00 | 0.82 | 7.58 | 18.57 | 1.00 | 0.98 | 6.23 | 5.01 | 11.24 |
| UCI Adult | 1 | 0.24 | 0.24 | 1.89 | 0.00 | 0.23 | 0.23 | 2.09 | 0.00 | 0.35 | 0.35 | 1.86 | 0.00 | 3.21 |
| | 2 | 1.00 | 0.62 | 1.94 | 0.86 | 0.89 | 0.56 | 4.53 | 1.98 | 1.00 | 0.67 | 2.06 | 0.85 | 4.50 |
| | 3 | 1.00 | 0.54 | 2.39 | 1.37 | 0.89 | 0.47 | 4.93 | 2.09 | 1.00 | 0.56 | 2.40 | 1.38 | 4.01 |
| | 4 | 1.00 | 0.48 | 2.35 | 2.12 | 0.89 | 0.39 | 4.91 | 3.15 | 1.00 | 0.63 | 2.37 | 2.11 | 4.03 |
| | 5 | 1.00 | 0.58 | 2.28 | 2.09 | 0.90 | 0.44 | 4.20 | 3.05 | 1.00 | 0.63 | 2.33 | 2.07 | 5.34 |
| | 6 | 1.00 | 0.56 | 2.33 | 2.16 | 0.91 | 0.44 | 3.75 | 2.99 | 1.00 | 0.60 | 2.37 | 2.15 | 3.58 |
| Max Error % | | 6.23 | 8.28 | 31.71 | 31.71 | 3.21 | 8.45 | 12.86 | 21.53 | 2.89 | 2.89 | 7.72 | 14.83 | |

Table 6: Number of clusters for k-means clustering results. Metrics are computed on scaled data and averaged over 10 trials. Maximum standard error bounds for each metric by task and across tasks are included.

Me Some Credit in Table 2. However, the other clusters were reachable for most PoIs, meaning StEP almost always had a successful recourse path for every PoI.

From our empirical results, the consideration around number of clusters has more to do with how the clusters are distributed in high-dimensional space relative to the learned decision boundary of each classifier, rather than as a function of "cluster goodness" or quality. From these results, we show that StEP is robust to the number of clusters for k-means, and therefore also robust to the relative size of each cluster (Table 6).

We also generated results with uniformly random cluster assignments, i.e., each point is randomly assigned a cluster value from $\{1, ..., k\}$ (Table 7). The $k = 1$ case for both k-means and random clustering produce the same results (as expected). For random clustering, as $k$ increases, we observe small improvements across our metrics of interest and that StEP demonstrates reasonable performance on most tasks. We note that, given the strict categorical constraints in the UCI Adult task, some clusters are inherently less reachable for many PoIs.

## B.7 Additional StEP User-Interference Results

In Table 5 present results on the path length, path steps, and proximal diversity metrics (defined in Section B.2). The noise $\beta$ we introduce offers insights into StEP's behavior. Interestingly, introducing noise produces improvements in performance on the Give Me Some Credit Dataset (Table 3) by decreasin path length and path steps while providing a small boost in proximal diversity. Recall that StEP clusters the *positively* labeled evaluation data which introduces additional hyperparameters: the number of clusters, $k$, and the clustering method. We cluster the evaluation data in full *without removing any outliers* via sci-kit learn's off-the-shelf k-means algorithm. In particular, one cluster in the Give Me Some Credit dataset contained several outliers.

| Dataset | $k$ | Logistic Regression | | | | Random Forest | | | | DNN | | | | Max Error % |
|---|---|---|---|---|---|---|---|---|---|---|---|---|---|---|
| | | Success | Avg Success | $\ell_2$ Dist. | Diversity | Success | Avg Success | $\ell_2$ Dist. | Diversity | Success | Avg Success | $\ell_2$ Dist. | Diversity | |
| Credit Card Default | 1 | 0.90 | 0.90 | 5.03 | 0.00 | 1.00 | 1.00 | 2.99 | 0.00 | 1.00 | 1.00 | 4.65 | 0.00 | 1.27 |
| | 2 | 0.90 | 0.89 | 5.04 | 0.05 | 1.00 | 1.00 | 2.99 | 0.03 | 1.00 | 1.00 | 4.65 | 0.01 | 10.48 |
| | 3 | 0.91 | 0.89 | 5.04 | 0.09 | 1.00 | 1.00 | 2.99 | 0.05 | 1.00 | 1.00 | 4.65 | 0.02 | 8.33 |
| | 4 | 0.92 | 0.89 | 5.04 | 0.13 | 1.00 | 1.00 | 2.99 | 0.06 | 1.00 | 1.00 | 4.65 | 0.03 | 6.03 |
| | 5 | 0.93 | 0.90 | 5.05 | 0.15 | 1.00 | 1.00 | 2.99 | 0.07 | 1.00 | 1.00 | 4.65 | 0.04 | 6.10 |
| | 6 | 0.94 | 0.90 | 5.06 | 0.18 | 1.00 | 1.00 | 2.99 | 0.09 | 1.00 | 1.00 | 4.65 | 0.04 | 6.53 |
| Give Me Some Credit | 1 | 0.58 | 0.58 | 2.37 | 0.00 | 1.00 | 1.00 | 2.83 | 0.00 | 1.00 | 1.00 | 4.11 | 0.00 | 8.67 |
| | 2 | 0.59 | 0.58 | 2.37 | 0.03 | 1.00 | 1.00 | 2.82 | 0.04 | 1.00 | 1.00 | 4.11 | 0.00 | 32.44 |
| | 3 | 0.60 | 0.58 | 2.39 | 0.05 | 1.00 | 1.00 | 2.81 | 0.06 | 1.00 | 1.00 | 4.11 | 0.01 | 31.38 |
| | 4 | 0.62 | 0.59 | 2.39 | 0.07 | 1.00 | 1.00 | 2.82 | 0.07 | 1.00 | 1.00 | 4.11 | 0.01 | 30.58 |
| | 5 | 0.63 | 0.59 | 2.40 | 0.08 | 1.00 | 1.00 | 2.83 | 0.09 | 1.00 | 1.00 | 4.11 | 0.01 | 24.86 |
| | 6 | 0.66 | 0.59 | 2.40 | 0.09 | 1.00 | 1.00 | 2.80 | 0.10 | 1.00 | 1.00 | 4.11 | 0.01 | 25.59 |
| UCI Adult | 1 | 0.24 | 0.24 | 1.89 | 0.00 | 0.23 | 0.23 | 2.09 | 0.00 | 0.35 | 0.35 | 1.86 | 0.00 | 3.21 |
| | 2 | 0.25 | 0.25 | 1.90 | 0.08 | 0.23 | 0.22 | 2.10 | 0.06 | 0.36 | 0.35 | 1.87 | 0.08 | 22.89 |
| | 3 | 0.26 | 0.25 | 1.92 | 0.17 | 0.25 | 0.22 | 2.12 | 0.11 | 0.36 | 0.35 | 1.87 | 0.16 | 9.92 |
| | 4 | 0.27 | 0.25 | 1.93 | 0.20 | 0.25 | 0.22 | 2.13 | 0.16 | 0.37 | 0.35 | 1.89 | 0.20 | 11.87 |
| | 5 | 0.28 | 0.25 | 1.94 | 0.28 | 0.26 | 0.22 | 2.13 | 0.19 | 0.37 | 0.35 | 1.90 | 0.27 | 9.85 |
| | 6 | 0.28 | 0.25 | 1.95 | 0.31 | 0.26 | 0.22 | 2.15 | 0.23 | 0.37 | 0.35 | 1.90 | 0.26 | 7.62 |
| Max Error % | | 2,91 | 2.52 | 1.19 | 15.89 | 3.21 | 3.21 | 8.99 | 32.44 | 3.30 | 2.95 | 4.76 | 22.89 | |

Table 7: Number of clusters for random cluster assignment results. Metrics are computed on scaled data and averaged over 10 trials. Maximum standard error bounds for each metric by task and across tasks are included.

When generating recourse for a PoI $\vec{x}$, StEP produces a direction towards the centers of each of the $k$ clusters. Any outliers in the evaluation data can cause dramatic shifts in the directions, due to the sparse topology of the outlier clusters.

The Give Me Some Credit dataset exhibits significant variance between the mean path length across the clusters. Beyond removing outliers from the dataset, we suggest the following to correct for the variance influenced by the presence of outliers: (a) increase the number of clusters used and ignore the paths computed for a cluster which has an average path length of $\geq 2$ standard deviations higher than the mean path length between the other clusters; (b) increasing the number of clusters may help disperse the cluster assignments of outliers, and the more clusters StEP uses results in more paths produced, which can help support generating a greater number of viable suggested recourse options for a given stakeholder.

# C   Experimental Details for Reproducibility

## C.1   Datasets Details

**Credit Card Default:** we produce random train, validation, and test sets from the $30{,}000$ instances using a 70/15/15 split, resulting in sets with approximately 21k/4.5k/4.5k datapoints respectively. This dataset contains 24 features, 3 of which are categorical features. We also convert the columns denoted by "PAY$_*$" to continuous values by replacing instances of $-1, -2$ with 0. We set "MARRIAGE", "AGE", and "SEX" as immutable features. For StEP we set "EDUCATION" as an ordinal feature that can only increase.

**Give Me Some Credit:** we produce randomly assigned train, validation, and test splits from the instances in the same manner as with Credit Card Default. Give Me Some Credit consists of 10 features. We remove

points in this dataset with missing values prior to splitting into train, validation and test sets. We set "age" as an immutable feature.

**UCI Adult/Census Income:** we produce randomly assigned train, validation, and test splits from the instances in the same manner as with Credit Card Default. UCI Adult/Census Income consists of 15 features, 6 of which are categorical. We drop the "education" feature as it is equivalent to the "education-num" feature. For StEP we set "education-num" as a feature that can only increase when giving recourse. We set the following features as immutable features: "age", "marital-status", "relationship", "race", "sex", "native-country".

## C.2 Base Model Details

Each method relies on a base machine learning model; in our experiments, we use sci-kit learn's implementation of logistic regression and random forest replicating practitioners' preferences for simpler and more explainable models. To evaluate more complex models, we use the PyTorch library implement a non-linear neural network model with two hidden layers of 16 and 32 neurons. We use simple holdout set validation to determine the hyperparameters used, outlined in C.4.

## C.3 StEP Parameters Discussion

**Choosing a Distance Metric:** We use the simple rotation invariant $\ell_2$-norm as our distance metric. To ensure the distance function is not biased towards any feature, we normalize continuous feature values with respect to their mean and standard deviation (Mothilal et al. (2020b) follow a similar methodology). More formally, for a continuous feature $i$, we transform its value as follows:

$$x_i^j := \frac{1}{\sigma_i}(x_i^j - \mu_i)$$

where $\mu_i$ is the mean value of the feature and $\sigma_i$ is the standard deviation.

**Choosing the $\alpha$ function** We propose two different $\alpha$ functions — the *volcano* function* and the *sloped* function. The volcano function weighs nearby points higher than faraway points, but all points closer than a specific threshold are weighed equally (a similar function is used by Patel et al. (2022)). We denote this function by $\alpha_v$ and define it as follows:

$$\alpha_v(z; d, \gamma) = \begin{cases} \frac{1}{z^d} & z > \gamma \\ \frac{1}{\gamma^d} & z \leq \gamma \end{cases} \tag{4}$$

The sloped function is shaped like a normal distribution curve with a 0 mean. We denote this function by $\alpha_s$ and define it as follows

$$\alpha_s(z; w) = \exp\left(-\frac{1}{2}\left(\frac{z}{w}\right)^2\right)$$

Both of these functions are continuous and upper bounded. If we divide the output of these functions by $z$ (the input to the functions), the output will satisfy the conditions of Theorem 3.4, i.e. the StEP algorithm can be made differentially private with a slightly modified version of these $\alpha$ functions.

In our experiments, we use the volcano function $\alpha_v$ with $d = 2$ and $\gamma = 0.5$.

## C.4 Other Hyperparameter Selection

We perform a basic grid search across base models, recourse methods, and datasets using sci-kit learn's GridSearchCV function. To determine appropriate hyperparameters to use across the base ML models and recourse methods, we roughly optimize for a reasonably high success rate and to minimize distance.

---

*so named for its shape when drawn on a whiteboard.

For each base ML model, we sweep over a confidence cutoff between $\{0.50, 0.55, 0.60, 0.65, 0.7\}$ and evaluate the performance of each recourse method. We find that 0.7 provides reasonable results across datasets, base models, and recourse methods while remaining a reasonable value a practitioner may use. We vary $k \in \{1, 2, 3, 4, 5\}$, the number of paths to produce for each PoI (and for StEP, the number of clusters to produce), and fix $k = 3$ for our comparative analysis and user-interference experiments.

For StEP, we consider step sizes in $\{0.10, 0.25, 0.50, 0.75, 1.00\}$ and fix a value of 1 across all experiments. FACE has two main hyperparameters: graph type in $\{k\text{-NN}, \epsilon, \text{KDE}\}$ and distance threshold (a float). In B.4, we provide substantial discussion on the distance threshold parameter. For all our experiments using FACE, we used the $\epsilon$ graph following a suggestion from the authors of the method. For C-CHVAE we set the step distance hyperparameter to 1. We detail this hyperparameter in B.1. For all applicable recourse methods, we allow a maximum of 50 iterations to produce $k$ counterfactual(s).

## C.5 Computational Resources Used

We used two machines for our empirical evaluation, including for base model training (i.e. logistic regression, random forest, neural network), for all recourse experiments, and post-processing of results to produce metrics. These machines contain:

- 8 CPU cores, 64GB RAM, NVIDIA 4070 GPU, and 2TB local SSD disk memory.

- 16 CPU cores 64GB RAM, NVIDIA 4080S GPU, and 4TB local SSD disk memory.

