# OpenReview forum: "Simple Steps to Success: A Method for Step-Based Counterfactual Explanations"
_TMLR — Accepted by TMLR_

### Review · Reviewer_aKSn · 2024-07-16

**Summary Of Contributions:**

The paper designes a direction-based algorithm for producing counterfactual explanations, named StEP. It is shown that the directions computed by StEP uniquely satisfy a set of desirable properties, as well as differentially privacy under some mild assumptions. Empirical evaluation is demonstrated to show its performance comparing with baselines, and show its robustness to user-interference.

**Audience:**

Yes

**Claims And Evidence:**

Yes

**Requested Changes:**

NA

**Strengths And Weaknesses:**

Strengths:
1. Well-organized presentation.
2. Extensive experimental evaluation.

Weaknesses:
1. The idea of the stepwise explainable paths and the corresponding proofs are straightfoward. In other word, the contribution of the paper is limited.

---

### Review · Reviewer_1UEM · 2024-07-16

**Summary Of Contributions:**

The paper presents StEP, a new model-agnostic and data-driven method to find counterfactual explanations for algorithmic recourse. StEP provides a set of “directions” instead of proper counterfactuals, to gradually steer the user towards a high-density area of the data manifold achieving the desired classification outcome. The authors show how StEP satisfies some theoretical properties that ensure good explanations. The authors also show how StEP (without clustering) is privacy-preserving. Lastly, the paper presents an experimental section where the efficacy of StEP is compared with other competitors in terms of accuracy, diversity and closeness using real-world datasets.

**Audience:**

Yes

**Broader Impact Concerns:**

Please see the "Requested changes" section above.

**Claims And Evidence:**

Yes

**Requested Changes:**

- Could the authors add a “Broader Impact”/“Limitation” statement highlighting the potential societal pitfalls of their approach and/or limitations?
- Could the authors rephrase or better justify the naturalness assumption of the properties in Section 3.1?
- Can the authors comment on the computational complexity of Algorithm 1 and/or on its execution with limited data $\mathcal{X}$?
- What if the clustering used by StEP is not precise? Can the authors provide some theoretical/experimental results on this (e.g., by considering how StEP recourse success varies compared with some clustering evaluation metrics such as silhouette score/mutual information)?
- Can the author comment on the privacy-preserving properties of StEP (with clustering) in Section 3.3?

**Strengths And Weaknesses:**

The paper is well-written and well-organized. It is possible to understand mostly all the details and assumptions made, and also the proposed method is straightforward. The method is also well-framed in the literature and it provides a novel interpretation of the algorithmic recourse problem. I also praise the addition of examples (e.g., Figures 1, 2 and 3) that help motivate the approach and give an immediate gist of the idea. The experiments are well-executed and back up almost all claims made by the authors.

I have some comments mainly on the justification of some theoretical assumptions, the potential societal impacts of StEP and the efficacy of StEP under certain conditions.

In Section 2, the author argues that their approach “[...] offloads some computation costs onto the stakeholder” since they provide only directions rather than full-fledged recourse options. The authors also argue that “Even if the resulting change does not facilitate a change in outcome [...] we can still offer additional directions until a desirable outcome is obtained”. It seems to me that there are two issues here:
- The burden of implementing recourse is all on the user side. StEP can only offer instructions such as “raise your income”, without specifying how much. Basically, the users do not know the $c$ value in $f(x + c\cdot d)$. Thus, it might be suboptimal from a user standpoint since the stakeholders lack the full specifications.
- How many interactions with StEP will the user have before obtaining the desired classification? For example, a user could have a hard time following StEP directions (e.g., because the clustering is poor), thus increasing a lot the interactions.

I would suggest the author add a “Limitations” or a “Broader Impact” statement considering the potential effect on the application of StEP in real-world scenarios

In Section 1.1, the phrase “StEP is the _only method_ [...] which satisfies a set of _natural properties_” overstates the contribution. I believe the authors do not adequately justify why the properties proposed in Section 3.1 should be considered natural for explanations. What is _natural_ in the first place? Section 3.1 describes properties from a mathematical point of view, which is somewhat justified by citing previous research. However, since the ultimate goal of this work is to provide useful explanations for the user, I do not see how those mathematical properties match properties considered _realistic_ by a human.

In Section 2.1, the authors state that DiCE can recommend “impossible” actions (e.g., paying a negative amount of money) to obtain recourse. Following the latest DiCE implementation [1], it is possible to specify actionability constraints, which would in principle avoid this situation. Since the authors state in Section 4.1 that StEP encodes constraints in the same way as DiCE does, I wonder if the proposed approach suffers from the same issue, or if the authors picked a specific example where the competitor fails.

Algorithm 1 described the StEP method, but the authors do not discuss its computation complexity and/or potential pitfalls of running it for large datasets $\mathcal{X}$. I would expect that, since we consider only positive points, in practice the complexity would be linear on the size of $\mathcal{X}$. However, other issues might arise. For example, the similar method FACE tends to be very memory-consuming (if we use an adjacency matrix to build the graph we get $O(|\mathcal{X}|^2)$ space complexity), thus it is usually run with a reduced dataset $\hat{\mathcal{X}} \subset \mathcal{X}$.

Section 3.2 provides some intuitions if StEP is run without clustering. However, I believe that cluster quality is an important assumption and a prerequisite for StEP to work in practice. I would have liked to see an ablation experiment testing empirically this assumption, even on synthetic data (e.g., more practically, it is easy to create synthetic tasks with varying clustering difficulty, for example, by using utils such as make_classification in scikit-learn [2]).

Section 3.3 provides results for a privacy-preserving StEP method in the case it is run without clustering. Since as Section 3.2 points out, there are pitfalls in running StEP with no clustered $\mathcal{X}$, I wonder what is the significance of these results, since in practice we would expect to use the clustered StEP anyway.

Here below are some additional minor nitpicking comments:
- In Section 2, why do you need that the counterfactual explanation $\mathbf{x} \notin \mathcal{X}$? In principle, any instance $\mathbf{x} \in \mathcal{X}$ where $h(\mathbf{x}) = +1$ would constitute a good recourse suggestion.
- Section 3.1 feels like you are providing some properties which are conveniently satisfied by your solution a-posteriori, instead of the other way around. I would suggest having Section 3.1 before Equation 1.
- Table 2 and Table 3 are not very readable. I would suggest the authors highlight the desired property (e.g., StEP provides more diverse/closer solutions) to help match what is written in Section 4.2 with the Tables.
- Table 2 and Table 3 results might be improved by showing the standard error/standard deviation for each entry. Reporting the max error does not tell us if the results are statistically significant. For example, StEP might provide closer solutions, but if the std is enormous, then StEP might be worse than other competitors.


[1] https://interpret.ml/DiCE/readme.html

[2]  https://scikit-learn.org/stable/modules/generated/sklearn.datasets.make_classification.html#sklearn.datasets.make_classification

---

### Review · Reviewer_buBt · 2024-07-29

**Summary Of Contributions:**

The paper introduces a data-driven method called StEP for computing recourses. For a specific user, StEP first divides the input dataset into clusters and identifies a 'direction' in the user’s feature space for each cluster. Users can enhance their features by following the suggested direction. StEP iteratively provides new directions based on the user's actions until the desired classification is achieved. These suggested directions are data-driven, can be made to have minimal dependence on the underlying model through adding privacy preserving noise to the input directions, and fulfill certain desirable properties for offering a recourse.


The paper does a good job at presenting a straightforward, model-agnostic algorithm for generating recourse, evaluated across multiple datasets and three popular model classes. However, there are areas for improvement, such as the limited evaluation of the robustness of generated recourse when noise is added to achieve privacy and when different clustering methods are used. Overall, the paper offers a solid foundation for future research in distance-based algorithmic recourse, though further enhancements would increase its comprehensiveness.

**Audience:**

Yes

**Broader Impact Concerns:**

I do not have concerns regarding the work's broader impact.

**Claims And Evidence:**

Yes

**Requested Changes:**

- [**Required**]: Analysing the impact of clustering methods on recourse accuracy: Since the recourse direction is highly dependent on the clustering algorithm, it would be helpful to see how different clustering methods impact the probability to obtain a recourse.
- [**Required**]: Analysing the impact of cluster size on recourse accuracy: Given a fixed clustering method, how do different values of the minimum cluster size impact the probability of obtaining a recourse? This experiment would help readers understand the sensitvity of the method to hyperparameters.
- [**Suggested addition**] *DP Motivation*: The work by [1] is a missing reference that seems highly relevant to this work. [1] has first shown that privacy concerns are relevant to recourse problems, and weaving this observation into this work would strengthen the motivation behind considering DP in recourse problems.
- [**Optional**] *Analysing the impact of DP on recourse costs*: Does there exist a tradeoff between recourse costs and making recourse DP? I would imagine that the worst case recourse costs might increase due to additional noise being added to the recourse directions.


[1] Pawelczyk et al (2023), "On the Privacy Risks of Algorithmic Recourse", AISTATS, https://arxiv.org/abs/2211.05427

**Strengths And Weaknesses:**

**Strenghts**
- The paper presents a straightforward method for generating recourses, which contributes to its ease of implementation and understanding. The method is principled in the sense that it follows from a set of axioms the authors propose.

- The notion of direction-based recourse is original, and seems interesting given its simplicity.

**Weaknesses**
- The paper does not (empirically) study changes to fundamentally important hyperparameters on recourse accuracy and recourse costs. In particular, an analysis is missing which studies the sensitivity of recourse accuracy and recourse costs as the underlying clustering method is changed or the number of clusters $k$ is varied.

---

### Decision · Action_Editor_XcEQ · 2024-10-10

**Recommendation:** Accept as is

**Comment:**

The paper is very readable; the contribution is well motivated, overall sensible and sufficiently well supported by empirical evidence. Two reviewers out of three are in favor of accepting the manuscript.  The remaining reviewer is more negative, but I find their opinion is chiefly motivated by issues that do not necessarily align with TMLR's objectives.

This being said, I strongly encourage the authors to implement all clarifications requested by the reviewers for the camera ready.

**Audience:**

Yes.  Counterfactual explanations/algorithmic recourse are definitely of interest to TMLR's audience.

**Claims And Evidence:**

All reviewers agree that the message is sufficiently well supported by empirical
validation, especially in light of the additional clustering experiments
requested by the reviewers.  The results, while not groundbreaking, have an
element of novelty to them.